# Mass spectral characterization of primary emissions and implications in source apportionment of organic aerosol

Weiqi Xu[1,#], Yao He[1,2,#], Yanmei Qiu[1,2], Chun Chen[1,2], Conghui Xie[1,2], Lu Lei[1,2], Zhijie Li[1,2], Jiaxing Sun[1,2], Junyao Li[1,2], Pingqing Fu[3], Zifa Wang[1,2], Douglas R. Worsnop[4], and Yele Sun[1,2,5,*]

[1]State Key Laboratory of Atmospheric Boundary Layer Physics and Atmospheric Chemistry, Institute of Atmospheric Physics, Chinese Academy of Sciences, Beijing 100029, China
[2]University of Chinese Academy of Sciences, Beijing 100049, China
[3]Institute of Surface-Earth System Science, Tianjin University, Tianjin 300072, China
[4]Aerodyne Research Inc., Billerica, Massachusetts 01821, USA
[5]Collaborative Innovation Center on Forecast and Evaluation of Meteorological Disasters, Nanjing University of Information Science & Technology, Nanjing 210044, China

#These authors contributed equally

*Correspondence*: Yele Sun (sunyele@mail.iap.ac.cn)

**Abstract.** Source apportionment of organic aerosol (OA) from aerosol mass spectrometer (AMS) or aerosol chemical speciation monitor (ACSM) measurements relies largely upon mass spectral profiles from different source emissions. However, the changes in mass spectra of primary emissions from AMS/ACSM with the newly developed capture vaporizer (CV) are poorly understood. Here we conducted 21 cooking, crop straw, wood, and coal burning experiments to characterize the mass spectral features of OA and water-soluble OA (WSOA) using SV-AMS and CV-ACSM. Our results show overall similar spectral characteristics between SV-AMS and CV-ACSM for different primary emissions despite additional thermal decomposition in CV, and the previous spectral features for diagnostic of primary OA factors are generally well retained. However, the mass spectral differences between OA and WSOA can be substantial for both SV-AMS and CV-ACSM. The changes in $f_{55}$ (fraction of $m/z$ 55 in OA) vs. $f_{57}$, $f_{44}$ vs. $f_{60}$, $f_{44}$ vs. $f_{43}$ in CV-ACSM are also observed, yet the evolving trends are similar to those of SV-AMS. By applying the source spectral profiles to a winter CV-ACSM study at a highly polluted rural site in North China Plain, the source apportionment of primary OA was much improved highlighting the two most important primary sources of biomass burning and coal combustion (32% and 21%). Considering the rapidly increasing deployments of CV-ACSM and WSOA studies worldwide, the mass spectral characterization has significant implications by providing essential constrains for more accurate source apportionment, and making better strategies for air pollution control in regions with diverse primary emissions.

# 1 Introduction

Organic aerosol (OA) is ubiquitous in the atmosphere and often contributes a large fraction of aerosol particles. Currently, Aerodyne aerosol mass spectrometer (AMS) is one of the most widely used instruments for real-time measurements of OA (Canagaratna et al., 2007;Li et al., 2017). OA can be further separated into primary OA (POA) and secondary OA (SOA)

factors by using receptor models, e.g., positive matrix factorization (PMF) and multilinear engine (ME-2) (Paatero, 1999;Paatero and Tapper, 1994). The determination of OA factors relies strongly upon the comparisons with collocated measurements and also the mass spectral profiles of primary emissions. However, in the absence of collocated measurements, the spectral features become the most important constrain for selection of PMF factors. As a result, the mass spectra of primary emissions have been extensively characterized with quadrupole- and high-resolution time-of-flight AMS, including traffic

exhaust (Canagaratna et al., 2004;Collier et al., 2015), biomass burning (Schneider et al., 2006;Alfarra et al., 2007) and cooking emissions (Mohr et al., 2009;He et al., 2010;Allan et al., 2010;Robinson et al., 2018), and the spectral characteristics, e.g., hydrocarbon ion series $C_nH_{2n-1}^+$ and $C_nH_{2n+1}^+$ for traffic emissions, $f_{60}$ (fraction of $m/z$ 60 in OA) for biomass burning, and high $f_{55}/f_{57}$ for cooking OA, are widely used as diagnostics for the presence of OA factors. However, coal combustion emissions, one of the most important primary sources in north China in winter, are rarely characterized (Lin et al., 2017). Owing to the

relatively similar spectra between coal combustion OA (CCOA) and traffic-related hydrocarbon-like OA (HOA), and the decreases in coal combustion emissions in Beijing in recent years, it becomes more challenging to separate these two fossil-fuel-related factors (Sun et al., 2016a;Xu et al., 2019), particularly for unit mass resolution spectra measured by aerosol chemical speciation monitor (ACSM). Therefore, it is of great importance to characterize the spectral features of CCOA for a better quantification of coal combustion emissions.

Although AMS/ACSM is capable of measuring OA in real-time, the uncertainties in quantification can be up to 38% (Bahreini et al., 2009) mainly due to the influence of collection efficiency (CE) caused by particle bouncing from the vaporizer (Huffman et al., 2005;Matthew et al., 2008). While the parameterization of CE as a function of particle phase, acidity, and the fraction of ammonium nitrate (Middlebrook et al., 2012) has improved the AMS/ACSM quantification, the applications for the $PM_{2.5}$ lens and particles larger than 1 μm remain unknown. As a result, a new capture vaporizer (CV) with an enclosed cavity was

developed (Xu et al., 2017b). Field measurements showed that the CE of CV-AMS was fairly robust at ~1 (Hu et al., 2017). However, the OA mass spectra can have significant changes in CV by shifting towards smaller fragments compared to standard vaporizer (SV) due to additional thermal decomposition (Hu et al., 2018a;Hu et al., 2018b). To our knowledge, the mass spectral differences between CV and SV, and the mass spectral features of primary emissions in CV-AMS/ACSM have not been well understood yet. Hence, it is critically important to re-characterize the mass spectra of primary emissions in CV, and

to provide essential constrains for OA source apportionment from the rapidly increasing CV-ACSM measurements worldwide.

In recent years, water-soluble OA (WSOA) which plays an important role in affecting aerosol hygroscopicity and cloud condensation nuclei formation, has attracted an increasing attention(Bozzetti et al., 2017;Xu et al., 2017a;Qiu et al., 2019;Ye et al., 2017). Due to the challenges in real-time on-line measurements of WSOA, most previous studies focus on offline analysis of WSOA using SV-AMS. The results showed that oxygenated OA (OOA) and biomass burning OA (BBOA) are generally more water-soluble than other primary sources emissions, e.g., traffic and cooking (Mayol-Bracero et al., 2002;Daellenbach et al., 2016). Because of the different water-solubility of OA factors, the mass spectra of WSOA can be substantially different from the total OA which increases the difficulties in separation of the WSOA factors in source apportionment of WSOA. Although several studies tried to use ME-2 for a better source apportionment of less water-soluble components, e.g., HOA and cooking OA (COA), by using the ambient resolved spectra as constraints (Bozzetti et al., 2017;Daellenbach et al., 2016), it could introduce additional uncertainties when water-soluble HOA and COA were different from the constrained spectra. Unfortunately, mass spectral characterization of WSOA from different primary emissions using SV-AMS and CV-ACSM is extremely limited. Thus, there is an urgent need for characterization of the mass spectra of WSOA from different primary emissions, which has a great potential to improve the future source apportionment of WSOA.

In this work, we conducted 21 cooking and burning experiments to characterize the mass spectral features of OA and water-soluble OA from cooking emissions, crop straw burning, wood burning and coal combustion using SV-AMS and CV-ACSM. The mass spectra of OA and WSOA from CV-ACSM are compared with those of SV-AMS, and the changes in specific marker *m/z*'s for different primary sources are elucidated. In particular, we demonstrate the importance of applying mass spectra of primary emissions to receptor models for a better source apportionment of OA in a highly polluted environment with complex primary emissions.

**2 Experimental methods**

**2.1 Experimental set-up**

21 experiments were conducted including 7 cooking with different oils (stir-fried garlic with corn oil, stir-fried celery with corn oil, peanut oil, bean oil, sunflower oil, blend oil, and lard oil), one barbecue, 6 crop straw burning (dry wheat, corn, bean, rape, and cotton) and 4 wood burning (dry birchen, pine tree, poplar, and Chinese oak) under smoldering-dominated conditions, and 4 coal combustion (brown and bituminous coal) under both flaming and smoldering conditions in June 2019. The average ($\pm 1\sigma$) temperature and relative humidity during the experiments were 24.6 ($\pm 3.4$) °C and 59.5 ($\pm 23.1$) %. All fuels were burned in a common residential stove outside a 50 m$^3$ tent, and aerosol particles were then emitted into the tent through a chimney (Fig. 1). After approximately 5 min, a high volume sampler (TISCH) was first used to collect PM$_{2.5}$ samples for 10 min, then a HR-AMS equipped with an SV and PM$_1$ lens (SV-AMS hereafter) and a ToF-ACSM equipped with a CV and PM$_{2.5}$ lens

(CV-ACSM hereafter) were operated in parallel to measure organic aerosol particles for approximately 15 min. Because the real-time measurements of $CO_2$ were not available, a HEPA filter was placed in front of the sampling line before and after the SV-AMS and CV-ACSM measurements to correct the influence of gaseous $CO_2$ on the total $m/z$ 44. After the burning experiment, the room was ventilated completely until the mass concentrations of aerosol particles were close to the ambient values that were measured by the other CV-ACSM nearby.

Cooking experiments were conducted inside the tent by simulating the real Chinese cooking styles with different oils. To avoid the influences from burning of the fuel, an induction cooker was used in this study. In comparison, the barbecue experiment was performed using mutton shashlik and anthracite as ingredient and fuel, respectively, which are the most popular barbecue styles in restaurants. Burning anthracite alone was found to emit significantly lower mass loadings of aerosol particles than those emitted from barbecue.

The average mass loadings of OA during the burning and cooking experiments are nearly 2 order of magnitude of that in ambient air, indicating the negligible influences of background OA to our experiments. As shown in Table S1, the mass concentrations of OA measured by SV-AMS ranged from ~80 µg m$^{-3}$ to ~1370 µg m$^{-3}$ for different burning experiments by using a relative ionization efficiency of 1.4 and a collection efficiency of 1. Considering that the mass spectra of OA can have changes across different mass loadings due to the partitioning of semi-volatile organic compounds (Donahue et al., 2006; Shilling et al., 2009), we further checked the spectral differences between high and low mass loadings for SV-AMS and CV-ACSM (Tables S1 and S2, respectively). As indicated in Figures S1 and S2, the mass spectra of OA, and $f_{44}$, $f_{43}$, and $f_{60}$ from cooking and flaming combustion of coal are remarkably similar under low and high mass loadings, indicating that the mass spectra are relatively stable upon dilution or evaporation, and thus can be well used as constraints in source apportionment analysis. Although the mass spectra of OA for the rest burning, i.e., biomass burning, wood burning, and smoldering combustion of coal are also highly similar between low and high mass loadings, the ubiquitous increases in $f_{44}$ and corresponding decreases in $f_{60}$ were observed from high to low mass loadings. For instance, $f_{44}$ in SV-AMS was increased by 0.4 – 2% as the mass loading decreased by a factor of ~3, and $f_{60}$ showed a corresponding decrease by 0.1 – 0.9%. Similarly, $f_{44}$ in CV-ACSM was increased by 0.9 – 4.2% associated with a decrease in $f_{60}$ by 0.1 – 0.6% as OA mass loadings were decreased by a factor of ~3 – 4. Such results are consistent with previous studies that biomass burning OA can be rapidly aged in the atmosphere which is characterized by increases in $f_{44}$ and decreases in $f_{60}$ (Cubison et al., 2011; Morgan et al., 2020). Therefore, source apportionment of OA using the source spectra from biomass burning, wood burning and smoldering combustion of coal need to consider the mass loading effects and increase the variability uncertainties in $f_{44}$ and $f_{60}$.

## 2.2 Chemical and data analysis

The PIKA 1.57 and Tofware v2.5.13 were used for determination of mass concentrations and mass spectra of OA measured by SV-AMS and CV-ACSM, respectively. The elemental ratios of OA measured by SV-AMS including hydrogen-to-carbon (H/C), oxygen-to-carbon(O/C), nitrogen-to-carbon (N/C) and organic mass-to-organic carbon (OM/OC) ratios in this study were calculated using the Improved-Ambient (I-A) method (Canagaratna et al., 2015). The offline analysis of WSOA with SV-AMS and CV-ACSM is similar to that reported in our previous study (Qiu et al., 2019). Briefly, 2 or 3 punches of filter samples were sonicated in 25 mL deionized water, and then filtered with 0.45 μm syringe filters (Anpel, PVDF). An aliquot of the solution was atomized using pure argon, dried by the nafion dryer, and then simultaneously measured by SV-AMS and CV-ACSM. Different from WSOA in ambient aerosol (Qiu et al., 2019), the ratio of $CO^+/CO_2^+$ varied largely among different burning experiments, for example, 0.78 – 1.40 for cooking, 2.07 – 2.53 for crop straw burning, 1.50 – 2.45 for wood burning, and 1.50-1.85 for coal combustion (Fig. 2). For a better comparison with OA measured by SV-AMS, $CO^+$ was scaled to be equal to $CO_2^+$ for all WSOA samples. We found that the O/C ratios calculated with $CO^+ = CO_2^+$ and the fitted values have differences by 0.6 – 11.6%. In addition, organic carbon (OC), elemental carbon (EC), and water-soluble OC (WSOC) in $PM_{2.5}$ samples were analyzed by a Sunset OC/EC analyzer (Sunset Laboratory Inc., Model-4) and a total organic carbon (TOC) analyzer (Shimadzu, TOC-L), respectively. A more detailed description of carbonaceous aerosol analysis is given elsewhere (Li et al., 2018).

## 3 Results and discussion

### 3.1 Cooking emissions

The mass spectral profiles of 8 cooking OA are shown in Figs. 3-4 and S3. All COA spectral profiles measured by SV-AMS and CV-ACSM are highly similar ($R^2$>0.89, Fig. 5), and also resemble those previously resolved in ambient air during all seasons in Beijing (Fig. S4) despite the COA concentrations can have a difference of an order of magnitude. We also noticed slightly higher O/C ($f_{44}$) for COA under lower mass loadings, which were likely due to partitioning of more semi-volatile organics on particles during periods with higher mass loadings (Reyes-Villegas et al., 2018a). These results suggest fairly robust COA spectra for different cooking oils. One explanation is the relatively similar ingredients of cooking oils that are generally dominated by fatty acids and carbonyls (Schauer et al., 2002). Consistent with previous studies (Sun et al., 2011;Mohr et al., 2012), the source spectra of COA from SV-AMS are characterized by high $f_{55}/f_{57}$ (fraction of $m/z$ 55 and 57 in OA, respectively) ratios (2.0–2.7), and low O/C ratios (0.15 – 0.18). These results indicate that the COA source spectra can be used as good constraints for a better source apportionment of COA. We also observed considerable N-containing ions in COA spectra, and the average N/C ratios ranged from 0.005 to 0.033. Such high N/C ratios suggest that cooking emissions can be a

significant source of organic nitrogen (ON) in ambient air, in agreement with our previous study showing two ON peaks during mealtimes (Xu et al., 2017d), and also the ubiquitous identification of nitrogen-containing compounds from cooking emissions (Reyes-Villegas et al., 2018a). Note that the N/C ratio of ambient COA identified by PMF in Beijing (0.002-0.015) (Sun et al., 2016a;Xu et al., 2017c;Xu et al., 2019) is generally lower than those from cooking emissions (Fig. S5). One reason is due to

the challenges in separation and quantification of N-containing ions in ambient OA, particularly for the low mass resolution V-mode measurements. We also noticed pronounced $m/z$ 60 ($f_{60}$=0.57-0.96%) and $m/z$ 73 ($f_{73}$=0.59-1.1%) in COA source spectra, which are generally used as biomass burning tracers (Cubison et al., 2011). Because an induction cooker was used in this study, the signals of $f_{60}$ and $f_{73}$ would be completely from cooking oils. Previous studies also observed such signals from laboratory-generated cooking emissions, for example, palm oil COA (Liu et al., 2018;Liu et al., 2017), fresh COA

(Kaltsonoudis et al., 2017), heating of frying oil and deep-frying (Faber et al., 2013). Although the chemical ionization mass spectrometer was able to detect high concentrations of levoglucosan in cooking emissions (Reyes-Villegas et al., 2018a), the ratios of $f_{60}/f_{73}$ in COA from SV-AMS are fairly constant (~1, Fig. 6), which are approximately twice lower than those observed in biomass burning OA (~2, Fig. 6). These results highlight the contributions of other cooking-related oxygenated compounds to $m/z$ 60 and $m/z$ 73.

As shown in Figs. 3 and S6, all COA spectra of CV-ACSM are fairly stable and overall similar to those of SV-AMS ($R^2$> 0.86). Due to additional thermal decomposition in CV, the COA source spectra in CV showed slightly higher $f_{44}$ (2.4–3.7%) than that of SV-AMS (1.8–2.9%)(Hu et al., 2018a). The major COA spectral differences between CV-ACSM and SV-AMS are the changes in $C_nH_{2n-1}^+/C_nH_{2n+1}^+$ ratios, e.g., $m/z$ 41/43, $m/z$ 55/57, and $m/z$ 67/69. For example, the $m/z$ 55/57 ratios ranged from 2.8 to 5.4 in CV-ACSM, , which were consistent with those of cooking exhaust near a kitchen ventilator (4.05) measured by

another similar CV-ACSM (Zheng et al., 2020), yet the ratios were approximately twice higher than those in SV-AMS (2.0 – 2.7, Fig. 6). Similarly, the ratios of $m/z$ 41/43 and $m/z$ 67/69 in CV-ACSM (1.2 – 1.9 and 1.2 – 2.6, respectively) are also much higher than those in SV-AMS. In addition, we found that the prominent $m/z$ 60 and 73 signals in CV-ACSM were much smaller than those in SV-AMS, likely due to additional thermal decomposition in CV.

The mass spectra of water-soluble COA (WSCOA) are much different from those of total COA for both SV-AMS and CV-

ACSM in terms of elemental composition and $f_{44}$. As indicated in Figs. 4 and S3, the O/C ratios of WSCOA range from 0.33 to 0.45, which are much higher than 0.15-0.18 of COA, suggesting that WSCOA contains more oxygenated organic compounds. This is consistent with the much higher $C_xH_yO^+$ and $C_xH_yO_z^+$ families (23.4 – 34.5% and 10.0 – 12.5%, respectively) in WSCOA than COA (14.8 – 18.0% and 6.9 – 7.9%, respectively). Similarly, the $f_{44}$ of WSCOA is higher than that in COA by more than a factor of 2. We also noticed much higher N/C ratios in WSCOA than COA, indicating enriched nitrogen-containing

organic compounds in water-soluble COA. Despite the differences above, the mass spectral features of COA, i.e., high $m/z$ 55 and 57, and $m/z$ 55/57 ratio are well retained in WSCOA from animal oil for both SV-AMS and CV-ACSM (Fig. 4), while

there are more changes in WSCOA from vegetable oil with much reduced $f_{55}$ and $f_{57}$. One reason is due to the different water solubility of COA between vegetable and animal oils. Our carbon analysis showed that COA from vegetable oil has higher water-solubility compared to that from animal oil as indicated by the higher WSOC/OC ratios (~30% vs. 17%). As a result, the O/C ratio and $f_{44}$ in WSOA from vegetable oil are correspondingly higher than those from animal oil. By comparing with the source spectra of WSCOA, we found that the previously resolved ambient WSCOA in urban Beijing (O/C = 0.38 and WSCOA/OA = 19%) (Qiu et al., 2019) tends to be a mixture from cooking both vegetable and animal oils.

## 3.2 Crop straw burning

The mass spectral profiles of biomass burning have been relatively well characterized by SV-AMS in previous studies (Weimer et al., 2008;Lee et al., 2010;Schneider et al., 2006;Alfarra et al., 2007), and $m/z$ 60 (mainly $C_2H_4O_2^+$) and 73 (mainly $C_3H_5O_2^+$) from fragmentation of anhydrosugars (e.g., levoglucosan) are widely used as biomass burning markers in ambient studies. We found that $f_{60}$ measured by SV-AMS varied largely among different crop straw fuels ranging from 0.8% to 2.6% (Figs. 3 and S3) which is generally close to the values reported in previous studies (Sun et al., 2016b;Gilardoni et al., 2016), but much higher than that from open straw burning ($f_{60}$ = 0.3%-0.6%) (Fang et al., 2017). One explanation is that $f_{60}$ depends on biomass fuels, burning conditions (e.g., flaming or smoldering), and also chemical aging (Hennigan et al., 2011;Schneider et al., 2006;Collier et al., 2016). In addition, we also observed relatively high fractions of $C_xH_yN_z^+$ in BBOA (4.2% - 10.2%) and high N/C ratios (0.014-0.039), consistent with the observations of abundant nitrogen-containing organic compounds, e.g., N-heterocyclic alkaloid compounds, amines and nitrated phenols from biomass burning (Reyes-Villegas et al., 2018b;Bottenus et al., 2018;Wang et al., 2017;Laskin et al., 2009;Desyaterik et al., 2013). Compared with SV-AMS, the BBOA spectra of CV-ACSM are overall similar ($R^2$ = 0.93 – 0.96, Fig. S6) except the burning of wheat and corn stalk ($R^2$ = 0.80). As expected, $f_{60}$ in CV-ACSM is lower than that in SV-AMS (1.0% vs. 1.7%), while $f_{44}$ is correspondingly higher (5.9% vs. 2.7%) due to strong thermal decomposition in CV. Although the $f_{60}$ signal is low, it still can be used as a biomass burning marker for CV-AMS/ACSM (Hu et al., 2018a). It should be noted that the $f_{44}$ of crop straw burning measured by CV in this study is lower than that identified in ambient aerosol by the ToF-ACSM in Gucheng, likely indicating that ambient BBOA has been photochemically aged to some extent.

The mass spectra of water-soluble BBOA (WSBBOA) resemble those of BBOA for both SV-AMS and CV-ACSM (Figs. 4 and S3). One reason is due to the high solubility of BBOA of which ~40 – 70% of carbon was found to be water-soluble. This is consistent with the observation from a combustion chamber experiment (65%) (Zheng et al., 2020). It should be noted that the $f_{60}$ of WSBBOA measured by CV-ACSM in this study is higher than that reported in Zheng et al. (2020) likely due to the differences in combustion system and ACSM detectors. WSBBOA presents generally higher $f_{60}$ and $f_{44}$ than the total BBOA for SV-AMS (2.4% vs. 1.7%, and 4.1% vs. 2.7% respectively), and CV-ACSM (1.6 vs. 1.0%, and 8.7 vs. 5.9% respectively).

As shown in Fig. S7, the slope of $f_{44}$ is less than 1 for both SV-AMS and CV-ACSM, and all data points are located in the right-bottom corner, in agreement with the higher O/C ratios of WSOA than OA. In fact, the O/C ratios of WSBBOA is approximately 50% higher than those of BBOA although they are still lower than that (O/C = 0.59) identified in winter in urban Beijing (Qiu et al., 2019). These results further suggest that WSOA contains more oxygenated organic compounds with higher oxidation degrees.

## 3.3 Wood burning

Crop straw burning contributes dominantly to BBOA in China during both harvest season and winter (Chen et al., 2017;Sun et al., 2016b), while wood burning is more important for domestic heating in European countries (Mohr et al., 2011;Alfarra et al., 2007). Here we found that the mass spectra of BBOA and wood burning OA (WBOA) show relatively similar features (Fig. 3), which are both characterized by the prominent signals of $m/z$ 60 and $m/z$ 73. The $f_{60}/f_{73}$ varies from 1 to 2 for WBOA and BBOA, which is larger than that in cooking emissions ($f_{60}/f_{73}$ =~1) and CCOA ($f_{60}/f_{73}$ < 1, Fig. 6). Compared to crop straw burning, WBOA of SV-AMS generally shows much higher $f_{60}$ (2.5 –5.7% vs. 0.8 – 2.6%) and $f_{73}$ (1.3-2.3 vs. 0.6 – 1.4%), and higher oxidation degree with higher $f_{44}$ (1.7-5.5% vs. 1.8-3.4%) and O/C (0.23-0.51 vs. 0.16 – 0.38). These results suggest that wood burning appears to produce more anhydrosugar compounds. Consistent with BBOA, WBOA of CV-ACSM shows much higher $f_{44}$, and slightly lower $f_{60}$ than those of SV-AMS (Figs. 3 and S3). However, considering the spectral similarities between BBOA and WBOA, it would be very challenging to separate the two different biomass burning OA based only on AMS or ACSM measurements.

The mass spectra of WBOA of CV-ACSM show highly similar characteristics to those of SV-AMS ($R^2$=0.88-0.94) although $f_{60}$ is slightly lower and $f_{44}$ is comparably higher. Similar to BBOA, the water-soluble WBOA is also characterized by prominent peaks of $m/z$ 60 and $m/z$ 73 (Figs. 3 and S3), and enriched in oxygenated and nitrogen-containing organic compounds (O/C = 0.36 – 0.54 and N/C = 0.016-0.076). In fact, a large fraction of OC from wood burning was found to be water-soluble (32 – 40%). The $f_{44}$ vs. $f_{60}$ plot has been widely used in both field and laboratory studies to characterize the aging of biomass burning OA (Cubison et al., 2011;Hennigan et al., 2011). Photochemical aging of BBOA can be rapid under typical ambient OH levels, e.g., $1 \times 10^6$ molecules cm$^{-3}$ (Hennigan et al., 2010), and BBOA evolves quickly from the right-bottom to left-top region in $f_{44}$ vs. $f_{60}$, which is characterized by an increase in $f_{44}$ and a corresponding decrease in $f_{60}$. Although $f_{60}$ and $f_{44}$ of CV-ACSM and water-soluble BBOA/WBOA have differences compared with those measured by SV-AMS, the evolving trends in $f_{44}$ vs. $f_{60}$ are similar (Fig. 6). These results suggest that $f_{44}$ vs. $f_{60}$ can also be used as a good diagnostic for chemical aging of biomass burning aerosol that are measured by the CV-AMS/ACSM. It should be noted that some previous studies also found large differences in $f_{60}$ between SV-AMS and CV-AMS. For example, Hu et al. (2018a) found that the $f_{60}$ of OA from CV-AMS was lower than that from SV-AMS by a factor of 5 during a period with significant BB impacts, yet the correlation was high ($R$ =

0.70). One explanation is that the thermal decomposition of OA in CV could vary among different instruments, but the aging trends are similar between SV- and CV-AMS.

### 3.4 Coal combustion

Coal combustion emission is one of the most important primary sources of OA in winter in north China (Sun et al., 2013). Although CCOA was resolved and quantified by SV-AMS in several previous studies (Sun et al., 2016a;Hu et al., 2013), it becomes more challenging to separate it from traffic-related HOA in megacities of China due to their relatively similar mass spectra and diurnal variations. By burning two different types of coals, i.e., brown and bituminous coals under flaming and smoldering conditions, we found that the differences in CCOA spectra of SV-AMS can be substantial under different burning conditions, while the spectra are relatively similar under smoldering conditions ($R^2 = 0.98$, Fig. 5). Consistent with previously resolved CCOA in ambient aerosol, the CCOA source spectra are all characterized by prominent hydrocarbons ions (for example, $m/z$ 41, $m/z$ 43, $m/z$ 55 and $m/z$ 57), and PAHs-related fragments, e.g., $m/z$ 152, $m/z$ 165, $m/z$ 178, $m/z$ 189, $m/z$ 202, $m/z$ 215, etc. It is interesting to note that the CCOA spectrum in Beijing (Sun et al., 2016a) resembles more that of flaming combustion of bituminous coal, while that observed at Changdao island in central eastern China (Hu et al., 2013) shows more similarity to that of smoldering combustion of bituminous coal. This is consistent with the fact that the bituminous coal accounted for ~78% of the total coal production according to the China Coal Industry Yearbook (Zhou et al., 2016), yet the CCOA emissions can be different in different areas due to different combustion conditions. We also noticed that the signals of $m/z$'s >150 contribute approximately ~40% of the total signal of CCOA measured by SV-AMS (Fig. S8), which is much higher than those in crop straw/wood burning and cooking emissions, suggesting that CCOA contains much higher fractions of high molecular weight organic compounds, e.g., PAHs. Therefore, source apportionment studies in regions with large influences of coal combustion emissions needs to be cautious. For example, PMF analysis of high-resolution mass spectra of OA by limiting $m/z$ to 150 could underestimate CCOA substantially. Considering that the contribution of CCOA to OA is ~20% during wintertime in Beijing (Sun et al., 2016a), the missed $m/z$'s > 150 in PMF analysis could cause an underestimation of coal combustion by ~8%.

Similar to COA and BBOA, the CCOA spectra of CV-ACSM resemble those of SV-AMS (Fig. S6), yet with much higher $f_{44}$. Although the thermal decomposition and the increased residence time in CV result in the larger molecular-weight fragments shifting towards smaller ions, the PAHs signals are well retained in the mass spectra of CV-ACSM (e.g., $m/z$ 152, $m/z$ 165, $m/z$ 178, $m/z$ 189, $m/z$ 202, $m/z$ 215) due to the stabilized chemical structures of PAHs that are very resistant to fragmentation after ionization (McLafferty and Turecek, 1993), consistent with the observations of PAHs from burning different types of coals (Zheng et al., 2020). These results indicate that PAHs can also be used as tracers to identify the coal combustion OA measured by CV-ACSM. We also observed the differences in $f_{44}$ produced from flaming and smoldering combustion. While the flaming

combustion of brown coal produced higher $f_{44}$ than smoldering for both SV-AMS and CV-ACSM (5.7% vs. 2.9%, and 3.2% vs. 1.7%, respectively), it was reversed for bituminous coal burning. Previous studies showed that the CCOA spectra evolved during the combustion process depending on the burning temperature and oxygen supplied (Wang et al., 2013). In addition, $f_{60}$ was only observed to be significant in smoldering combustion of bituminous coal (0.46%, Fig. S3), suggesting that coal combustion can be a potential source of levoglucosan. For instance, Yan et al. (2018) observed the emissions of levoglucosan from semi-anthracite burning while medium- and high-volatile bituminous coals appear to be negligible sources.

The O/C ratios of coal combustion OA are generally low ranging from 0.08 to 0.23, which are close to those (0.14 – 0.22) identified in ambient air in winter and spring (Xu et al., 2019;Sun et al., 2016a;Hu et al., 2013). The low O/C ratios suggest relatively low water solubility of CCOA. Indeed, the WSOC from the four different coal combustion on average accounted for 16 – 34% of the total OC with the flaming combustion generating more water-soluble organic compounds (24 – 34%). As a result, the O/C ratios of water-soluble CCOA are approximately twice that of CCOA, and consistently, the contributions of oxygenated ions (36.9 – 49.7%) are much higher. As shown in Fig. S3, the mass spectra of water-soluble CCOA are quite different from the total CCOA, especially for flaming combustion of bituminous coal ($R^2$=0.24). Although the signals of high $m/z$'s are largely decreased in the mass spectra of water-soluble CCOA, the spectral characteristics of PAHs (e.g., $m/z$ 115, 128, 139, 152, 165, 181 etc.) are still observed (Fig. S9). Because of the reduced signals for large $m/z$'s, the spectral correlations between water-soluble CCOA and other water-soluble primary OA are much elevated (Fig. 5), increasing the difficulties for the identification of water-soluble CCOA. For example, the spectrum of water-soluble CCOA identified in winter in our previous study was much different from those measured in this study (Qiu et al., 2019), and the O/C ratio (0.68) was also much higher. One reason is likely due to the aging of CCOA during the transport to Beijing because of the ban of coal burning in the city. The uncertainties in PMF analysis caused by the spectra could be another possibility.

**4 Conclusion and Implications**

Aerodyne AMS and ACSM have been widely used to measure OA worldwide, and the subsequent source apportionment of OA by PMF or ME-2 relies largely upon the mass spectral profiles of primary emissions. While the mass spectra of primary emissions, e.g., cooking, biomass burning, and traffic are relatively well characterized by SV-AMS, their behaviors in the newly developed CV-AMS/ACSM are poorly known. Considering the rapid increases in deployments of CV-ACSM and the studies in water-soluble OA worldwide, it is critical importance to further characterize the mass spectra of primary emissions with CV-ACSM for a better source apportionment of OA in the future. In addition, the mass spectra of OA from coal combustion emissions, one of the most important primary sources in winter in China, are rarely characterized. By measuring 21 different primary emissions with SV-AMS and CV-ACSM, the similar spectral characteristics of primary OA between CV-ACSM and SV-AMS are demonstrated, yet the changes in specific marker $m/z$'s (e.g., $f_{44}, f_{43}, f_{60}, f_{73}$, etc.) and $m/z$ ratios ($f_{55}/f_{57}$,

$f_{41}/f_{43}$, $f_{60}/f_{73}$, etc.) for source diagnostics are also observed due to additional thermal decomposition in CV. Among all primary emissions, we found that the COA spectrum is the most robust for both SV-AMS and CV-ACSM, and has no clear dependence on oil gradients. However, the spectral differences between water-soluble OA and the total OA can be substantial for both SV-AMS and CV-ACSM depending on water solubility which is in the order of BBOA > WBOA > COA > CCOA. Noted that the mass loadings of primary emissions in this experiment are much higher than those in ambient air, which could cause some differences in water solubility and subsequent spectral differences in WSOA. In addition, we found that mass spectra of WSOA from CV-ACSM are relatively well correlated among different primary emissions, highlighting the challenges in source apportionment of WSOA using CV-ACSM in the future.

We further demonstrate the importance of mass spectra of primary emissions for OA source apportionment in a field campaign that was conducted with a PM$_{2.5}$ CV-ACSM at a highly polluted rural site in North China Plain in winter (Kuang et al., 2020). Positive matrix factorization of OA was able to identify four primary OA factors including traffic-related HOA, COA, BBOA and CCOA, and one secondary OOA. We found that the average contributions of HOA and COA (19% and 16%, respectively) were much higher than expected because the rural site is far from urban areas (~15 km) and tends to have small influences from cooking and traffic emissions. Although the temporal variations appeared to be reasonable and were correlated with specific tracer species (e.g., BC and CO), the COA spectrum showed unrealistically higher $m/z$ 27 and $m/z$ 29, and BBOA and CCOA spectra showed much lower $f_{44}$ than those observed in source profiles (Fig. 7). Therefore, we reperformed the OA source apportionment with ME-2 by constraining four primary OA factors. Considering the chemical environment of sampling site, we used the average mass spectra of COA from vegetable oil, flaming combustion of bituminous coal, crop straw burning of corn, and HOA resolved in Rizhao city by the same CV-ACSM as in this study in September (Lei et al., 2019) as constrains in ME-2 analysis with a-value ranging from 0 to 0.3. As shown in Fig. 7, the ME-2 results showed much reduced HOA and COA contributions (8% and 5%, respectively) compared with those from PMF analysis. As a result, BBOA and CCOA became the two major primary sources at the rural site in North China Plain (32% and 21%, respectively), consistent with the fact that coal and crop straw are the major fuels for residential heating. Also, the unrealistically high morning COA peak in PMF analysis was disappeared in the ME-2 analysis further supporting the rationale of ME-2 results (Fig. 7). Although the average contributions of POA and SOA are very close with and without constraining POA spectral profiles, the apportionment of POA factors can be improved substantially. The accurate source apportionment results have significant implications for future air pollution mitigating strategies, for instance, our new results highlight that reducing coal combustion and biomass burning emissions would be the most effective measure to improve the winter air quality in rural areas in North China Plain.

*Data availability.* The data in this study are available from the authors upon request (sunyele@mail.iap.ac.cn).

*Author contributions.* YS, QW and YH designed the research. QW, YH, YQ, CC, LL, and ZL conducted the experiments. LL,

QW, YH and YQ analyzed the data. YJ, PF, ZW and DW reviewed and commented on the paper. QW, YH, and YS wrote the paper.

*Competing interests.* The authors declare that they have no conflict of interest.

*Financial support.* This work was supported by the National Natural Science Foundation of China (41975170, 91744207).

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

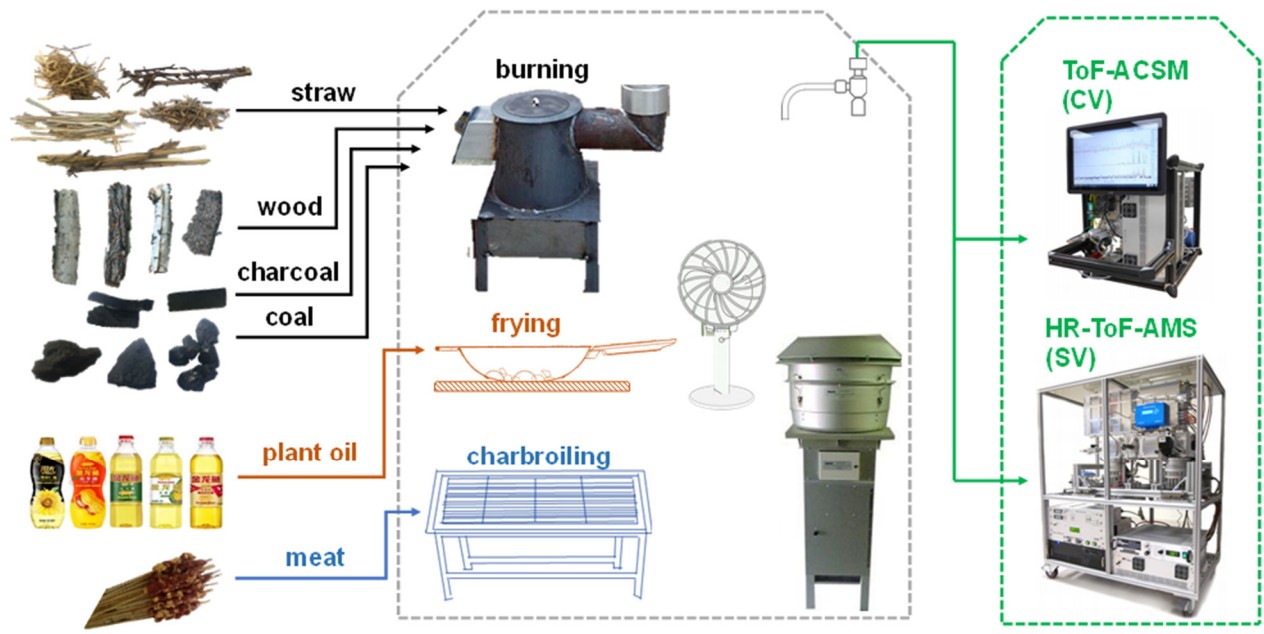

**Figure 1. Schematic of cooking and burning experiments.**

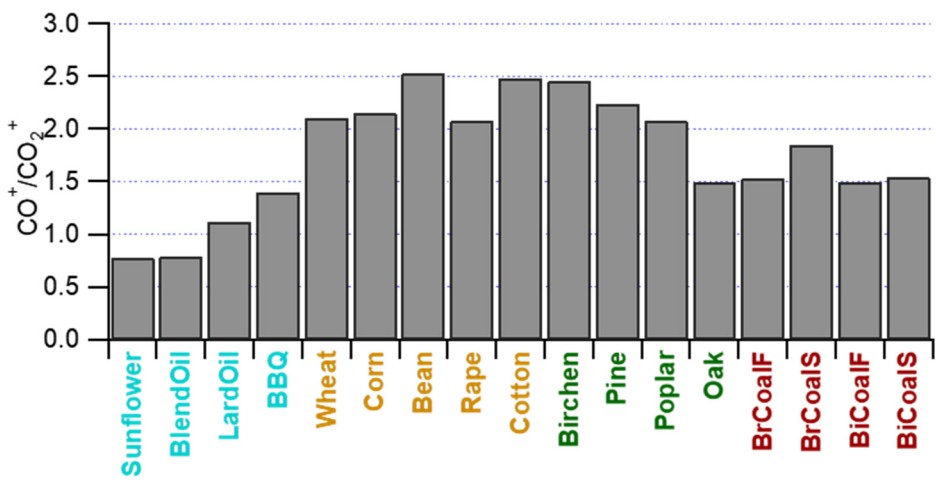

**Figure 2. The ratio of measured $CO^+/CO_2^+$ for WSOA from 17 cooking and burning experiments.**

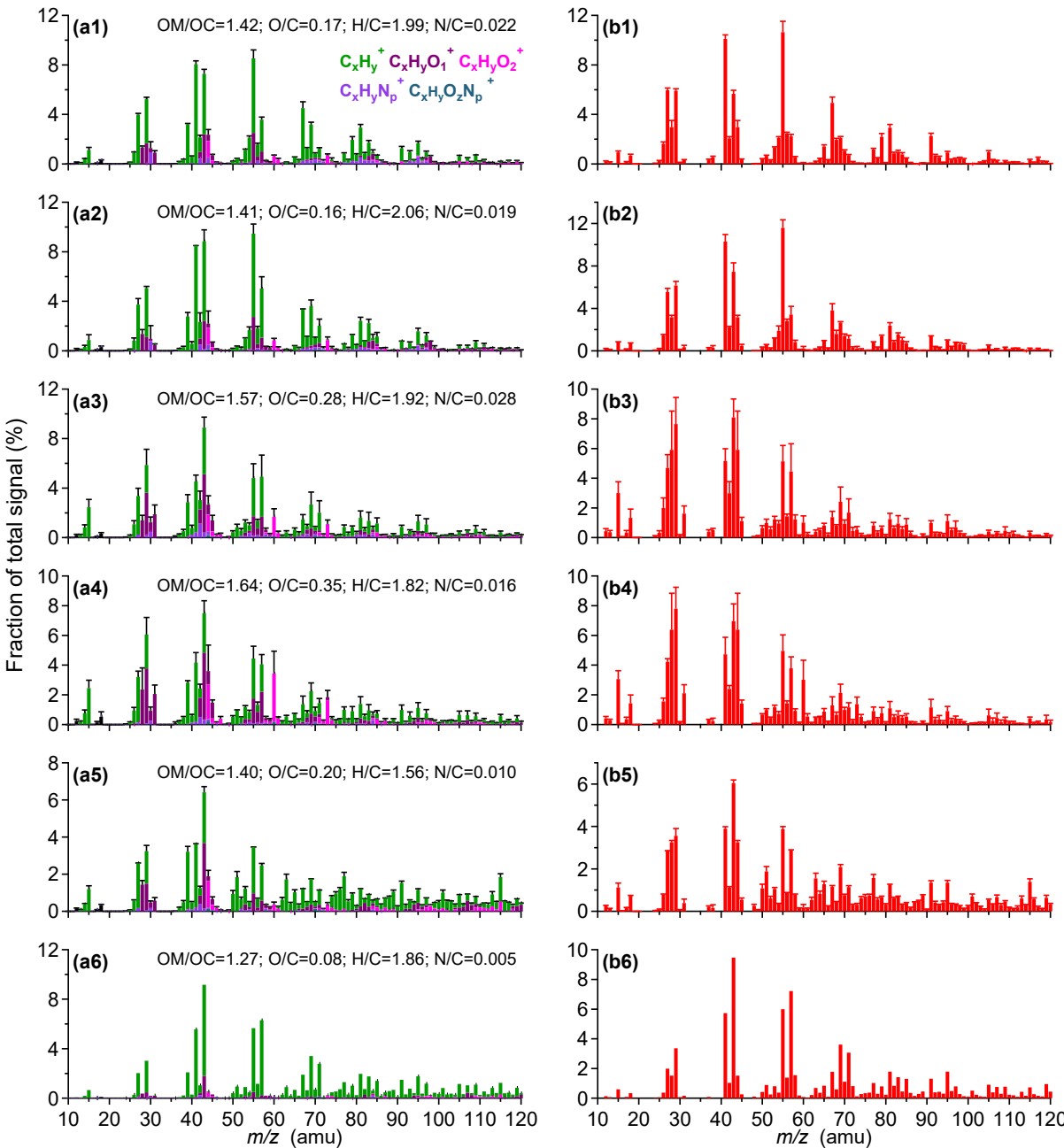

**Figure 3.** Average mass spectral profiles of OA measured by (a) SV-AMS and (b) CV-ACSM including (a1-d1) cooking emissions from vegetable oil, (a2-d2) from animal oil, (a3-d3) from crop straw burning, (a4-d4) from wood burning, and (a5-d5) and (a6-d6) from smoldering and flaming combustion of bituminous coal, respectively. The error bars represent one standard deviations. The elemental composition of OA and WSOA from SV-AMS are also shown. The detailed mass spectra for each cooking and burning are presented in Fig. S3.

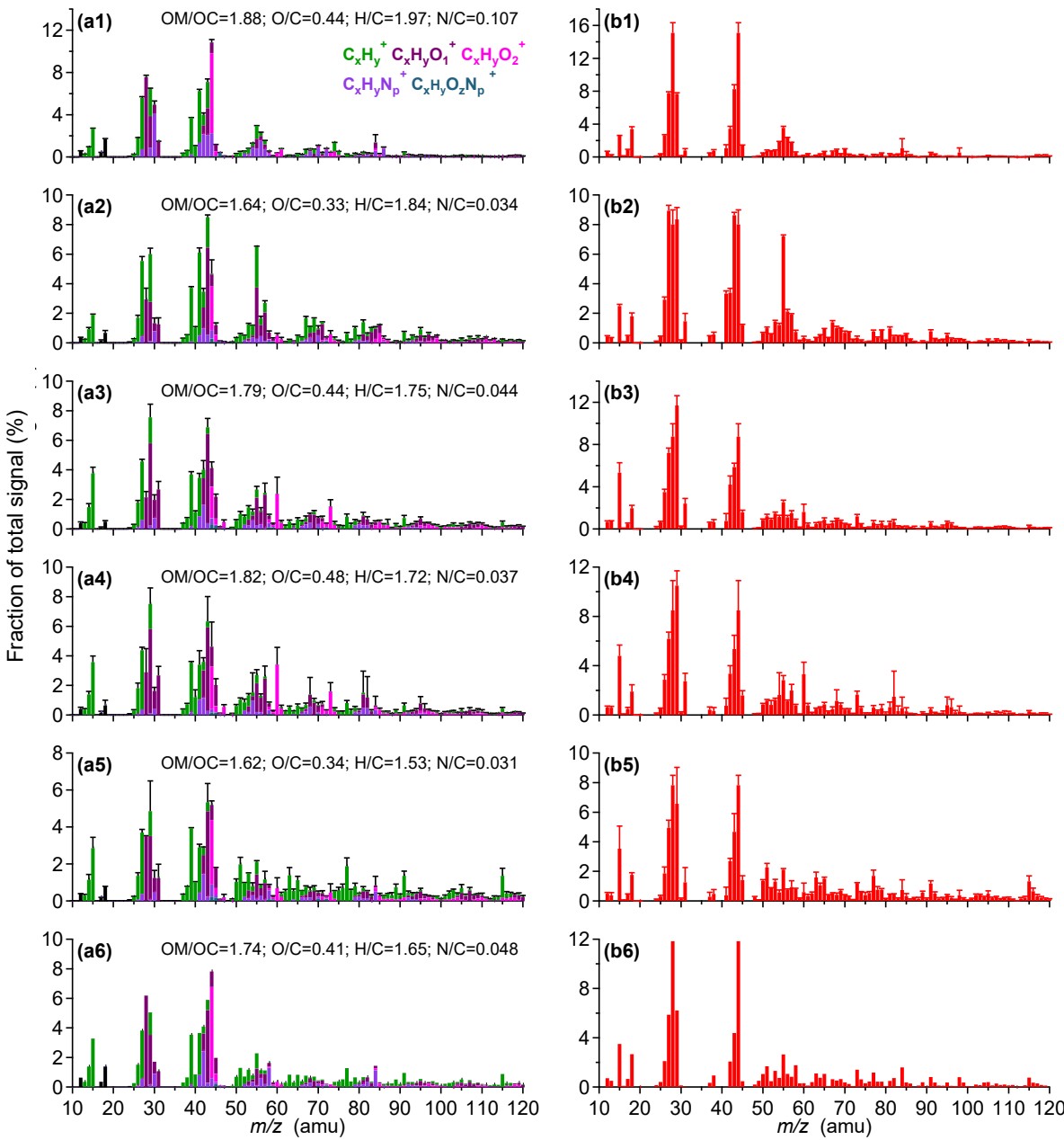

**Figure 4.** Average mass spectral profiles of WSOA measured by (a) SV-AMS and (b) CV-ACSM including (a1-d1) cooking emissions from vegetable oil, (a2-d2) from animal oil, (a3-d3) from crop straw burning, (a4-d4) from wood burning, and (a5-d5) and (a6-d6) from smoldering and flaming combustion of bituminous coal, respectively. The error bars represent one standard deviations. The elemental composition of OA and WSOA from SV-AMS are also shown. The detailed mass spectra for each cooking and burning are presented in Fig. S3.

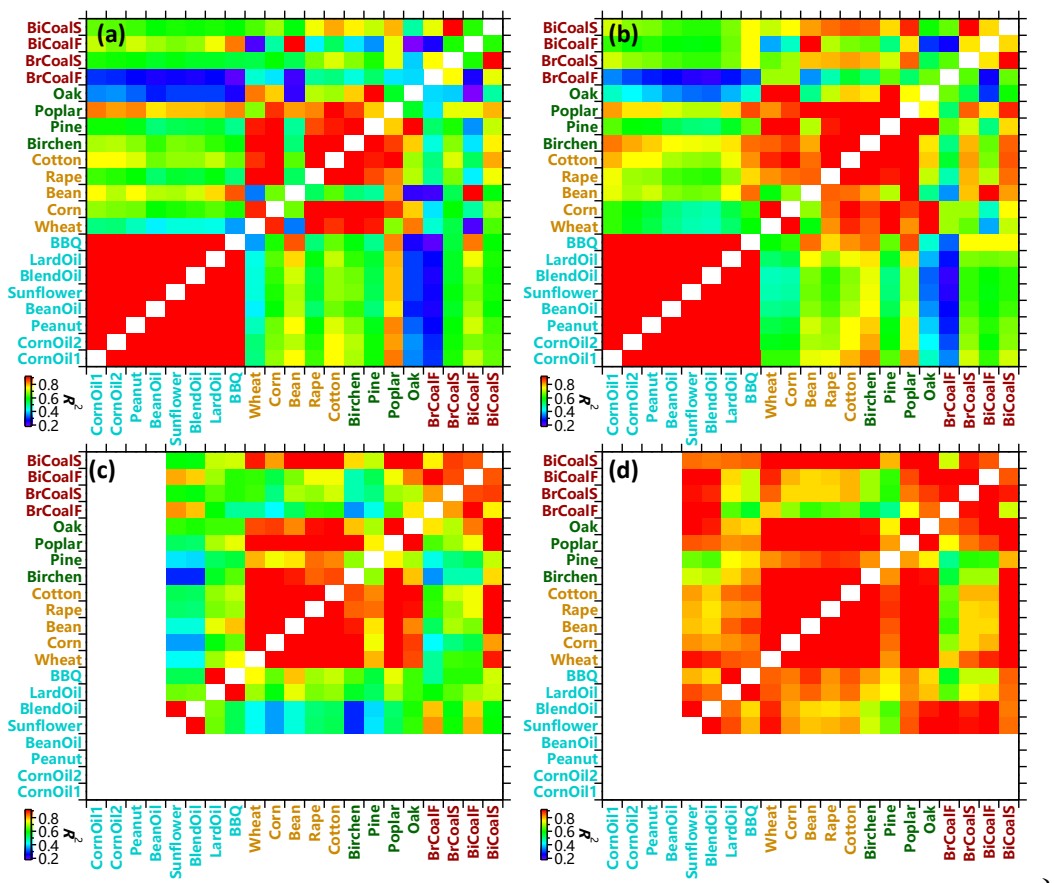

**Figure 5.** Mass spectral correlations of OA measured by (a) SV-AMS, (b) CV-ACSM, and WSOA measured by (c) SV-AMS and (d) CV-ACSM. The detailed descriptions of cooking and burning fuels are presented in Table S3.

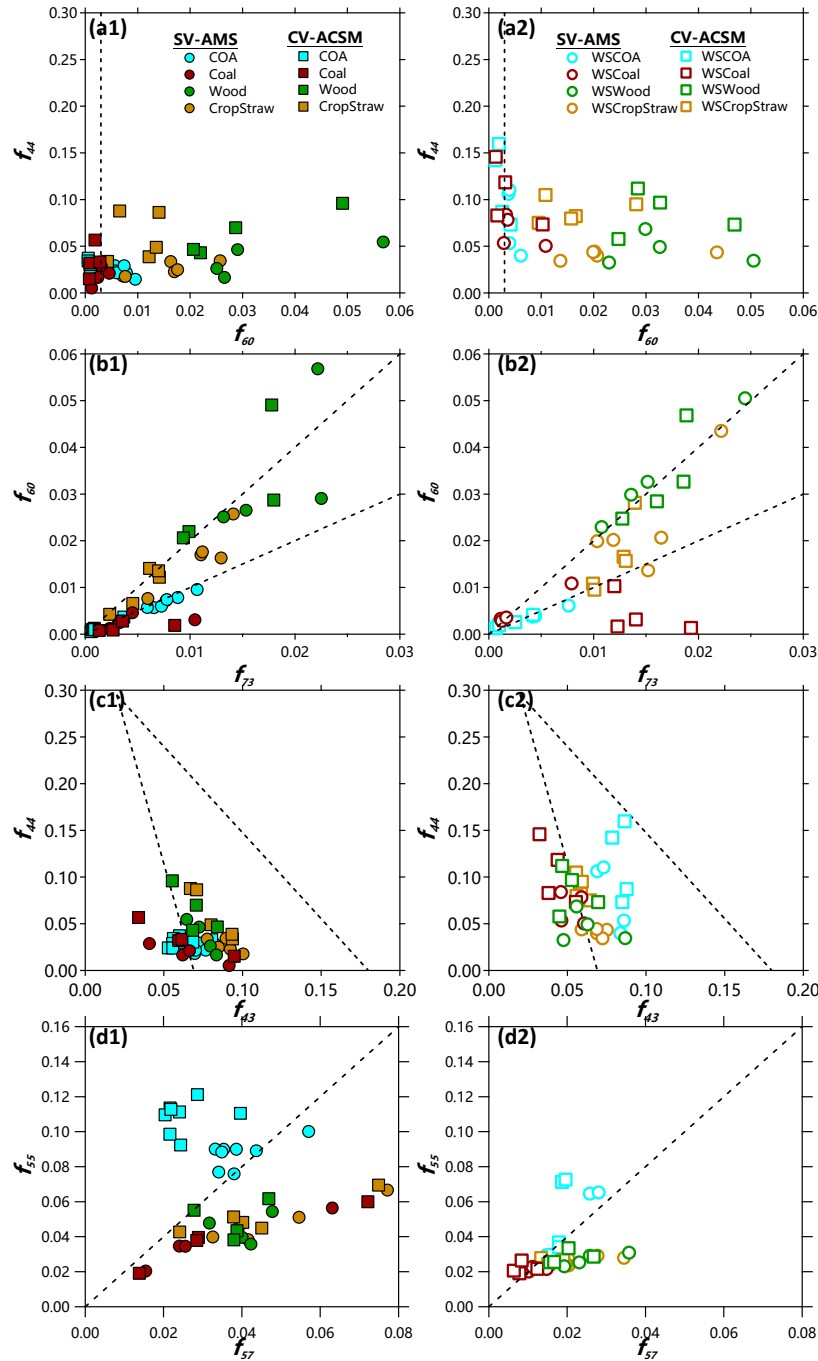

**Figure 6. Scatter plots of (a)** $f_{44}$ **vs.** $f_{60}$**, (b)** $f_{60}$ **vs.** $f_{73}$**, (c)** $f_{44}$ **vs.** $f_{43}$ **and (d)** $f_{55}$ **vs.** $f_{57}$ **for OA (top panel) and WSOA (bottom panel) from SV-AMS and CV-ACSM measurements.**

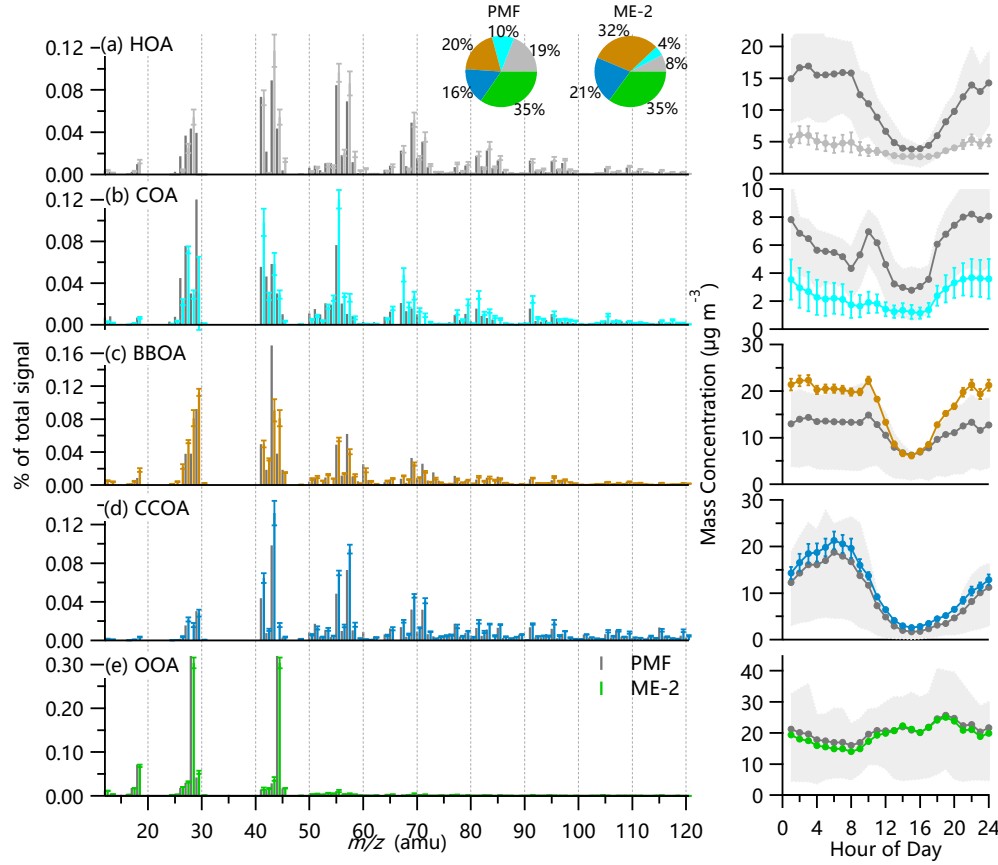

**Figure 7.** Comparisons of mass spectral profiles, diurnal cycles and compositions of five OA factors between PMF and ME-2 analysis at a rural site in North China Plain in winter. The ME-2 results are the averages with a-value ranging from 0 to 0.3, and the error bars are one standard deviations. The shaded areas in right panel indicate 25th and 75th percentiles of PMF results.