# Peer review of "Mass spectral characterization of primary emissions and implications in source apportionment of organic aerosol"

_Atmospheric Measurement Techniques, 2020_

## Referee Comment (RC1) · Anonymous Referee #1 · 6 Apr 2020

This is a very important and necessary piece of work comparing mass spectral profiles of different organic aerosol types comparing the 'standard' vs 'capture' vaporisers used in the AMS and ACSM. While it is acknowledged that there are differences between the two, an extensive comparison for different 'real world' aerosols is currently lacking. The experiments are appropriately and methodically performed and include both online and offline measurements, making these results applicable to both. This paper demonstrates the improvement to ME-2 source apportionment when these profiles are applied, showing this to be a very important technical contribution that will aid analysis in the future. While the aerosols sampled are undeniably focused on Chinese sources, given the number of these instruments in use in China currently, this will still be of

much use to the community and is firmly within scope for AMT. The work is appropriately and methodically performed and generally well written. I have only a couple of minor comments, but otherwise recommend publication.

Data availability: Given the scope for utilisation of this data, I would strongly encourage the mass spectral profiles to be hosted on a public archive. Traditionally, this has been the University of Colorado database. One would expect that future utilisation of these profiles by others will drive up the paper's citations, so it will be in the authors' interests to do so.

Consider placing figures S1 and S2 in the main article, as I think these are of sufficient interest that they should exist there.

Page 2, line 14: Should be 'owing' rather than 'owining'
* * *

---

## Referee Comment (RC2) · Anonymous Referee #2 · 7 Apr 2020

This manuscript provides a comprehensive data set for the characterization of primary OAs using the CV-ACSM compared to the SV-AMS. Similar spectral characteristics were found between the SV-AMS and CV-ACSM, and the latter showed additional thermal decomposition in the spectra. There is another paper on AMTD that addresses the similar topic (https://www.atmos-meas-tech-discuss.net/amt-2019-449/). This study should make comparisons to that one. Overall the paper is well written. I recommend acceptance for publication on AMT after minor revisions.

**Specific comments:**

Page 3, Line 27: What kind of stove was used?

Figure 1 is difficult to read especially for the standard deviations. I suggest to split Figure 1 to 2 graphs (one for OA and the other for WSOA) and enlarge the mass spectra.

Page 5, Line 16-17: Please indicate what numbers were shown in the parentheses? I think you mean f60 and f73. Has other COA studies also showed pronounced f60 and f73? Please compare. Also, what fuel was used for cooking? Is it possible that the signals of levoglucosan come from the burning of the fuel?

Page 5, Line 20-24: If the CV-ACSM sampled PM2.5 and the SV-AMS sampled PM1, there might be a composition difference. When comparing the two (not only for COA but also for other OAs), please justify the conclusions with that in mind.

Also, the authors mentioned about less enhancement of f44 compared to Hu et al. 2018a. Can this be partially explained by the loading difference? I mean the loadings herein were 2 orders of magnitude greater than ambient OA concentrations. More volatile species may partition to the particle phase compared to Hu et al. 2018a as well as the other study that I mentioned earlier. Please indicate the difference of conditions when making the comparisons.

Page 10, Line 6-7: The water-soluble fraction of POA also depends on the atmospheric dilution of the primary sources. This study should discuss about the sampling OA loading levels and the possible change of solubility after atmospheric dilution. Would that change the order of solubility?

The loading-dependent OA composition also limits the application of the source profile directly in ME-2 (Page 10, Line 18-25). For example, if the source profile is obtained at 1000 ug m-3, the actual source in the atmosphere is indeed tens of ug m-3 after quick dilution. Many studies have shown the OA composition varied a lot at various loadings especially for combustion sources like BBOA and CCOA and hence may change the mass spectra. The authors should be clarify this complication and do not mislead users to use the source profiles without cautions.

**Technical Remarks:**

"/" in "m/z" should not be italized.

---

## Author Comment (AC1) · 14 May 2020

We are thankful to the two referees for their thoughtful and constructive comments which help improve the manuscript substantially. Following the reviewers' suggestions, we have revised the manuscript accordingly. Listed below are our point-by-point responses in blue to each comment that is repeated in italic.

**Response to Reviewer #1**

*This is a very important and necessary piece of work comparing mass spectral profiles of different organic aerosol types comparing the 'standard' vs 'capture' vaporisers used in the AMS and ACSM. While it is acknowledged that there are differences between the two, an extensive comparison for different 'real world' aerosols is currently lacking. The experiments are appropriately and methodically performed and include both online and offline measurements, making these results applicable to both. This paper demonstrates the improvement to ME-2 source apportionment when these profiles are applied, showing this to be a very important technical contribution that will aid analysis in the future. While the aerosols sampled are undeniably focused on Chinese sources, given the number of these instruments in use in China currently, this will still be of much use to the community and is firmly within scope for AMT. The work is appropriately and methodically performed and generally well written. I have only a couple of minor comments, but otherwise recommend publication.*

We thank the reviewer's comments and have revised the manuscript accordingly.

*Data availability: Given the scope for utilisation of this data, I would strongly encourage the mass spectral profiles to be hosted on a public archive. Traditionally, this has been the University of Colorado database. One would expect that future utilisation of these profiles by others will drive up the paper's citations, so it will be in the authors' interests to do so.*

We will do that after the manuscript was accepted.

*Consider placing figures S1 and S2 in the main article, as I think these are of sufficient interest that they should exist there.*

It is a good point. We moved these two figures from supplementary to the main text in the revised manuscript.

*Page 2, line 14: Should be 'owing' rather than 'owining'*

Changed

**Response to Reviewer #2**

*This manuscript provides a comprehensive data set for the characterization of primary OAs using the CV-ACSM compared to the SV-AMS. Similar spectral characteristics were found between the SV-AMS and CV-ACSM, and the latter showed additional thermal decomposition in the spectra. There is another paper on AMTD that addresses the similar topic (https://www.atmos-meas-tech-discuss.net/amt-2019- 449/). This study should make comparisons to that one. Overall, the paper is well written. I recommend acceptance for publication on AMT after minor revisions.*

We appreciate the reviewer for pointing out this important paper. The comparisons between two studies have been made in the revised manuscript.

"For example, the $m/z$ 55/57 ratios ranged from 2.8 to 5.4 in CV-ACSM, which were consistent with those of cooking exhaust near a kitchen ventilator (4.05) measured by another similar CV-ACSM (Zheng et al., 2020), yet the ratios were approximately twice higher than those in SV-AMS (2.0 – 2.7, Fig. 6)."

"One reason is due to the high solubility of BBOA of which ~40 – 70% of carbon was found to be water-soluble. This is consistent with the observation from a combustion chamber experiment (65%) (Zheng et al., 2020). It should be noted that the $f_{60}$ of WSBBOA measured by CV-ACSM in this study is higher than that reported in Zheng et al. (2020) likely due to the differences in combustion system and ACSM detectors."

"the PAHs signals are well retained in the mass spectra of CV-ACSM (e.g., $m/z$ 152, $m/z$ 165, $m/z$ 178, $m/z$ 189, $m/z$ 202, $m/z$ 215) due to the stabilized chemical structures that are very resistant to fragmentation after ionization (McLafferty and Turecek, 1993), consistent with the observations of PAHs from burning different types of coals (Zheng et al., 2020)."

*Specific comments:*

*Page 3, Line 27: What kind of stove was used?*

The common residential stove was used in this study. We added the description in the revised manuscript to clarify it (also shown in Figure 1).

*Figure 1 is difficult to read especially for the standard deviations. I suggest to split Figure 1 to 2 graphs (one for OA and the other for WSOA) and enlarge the mass spectra.*

This is a good suggestion. We split Figure 1 into two figures in the revised manuscript.

*Page 5, Line 16-17: Please indicate what numbers were shown in the parentheses? I think you mean f60 and*

*f73. Has other COA studies also showed pronounced f60 and f73? Please compare. Also, what fuel was used for cooking? Is it possible that the signals of levoglucosan come from the burning of the fuel?*

Thank the reviewer's comments. The number in the parentheses is $f_{60}$ and $f_{73}$.

We expanded the discussions in the revised manuscript. Now it reads:

"We also noticed pronounced $m/z$ 60 ($f_{60}$=0.57-0.96%) and $m/z$ 73 ($f_{73}$=0.59-1.1%) in COA source spectra, which are generally used as biomass burning tracers (Cubison et al., 2011). Because an induction cooker was used in this study, the signals of $f_{60}$ and $f_{73}$ would be completely from cooking oils. Previous studies also observed such signals from laboratory-generated cooking emissions, for example, palm oil COA (Liu et al., 2018;Liu et al., 2017), fresh COA (Kaltsonoudis et al., 2017), heating of frying oil and deep-frying (Faber et al., 2013). Although the chemical ionization mass spectrometer was able to detect high concentrations of levoglucosan in cooking emissions (Reyes-Villegas et al., 2018a), the ratios of $f_{60}/f_{73}$ in COA from SV-AMS are fairly constant (~1, Fig. 6), which are approximately twice lower than those observed in biomass burning OA (~2, Fig. 6). These results highlight the contributions of other cooking-related oxygenated compounds to $m/z$ 60 and $m/z$ 73."

We described the cooking styles in section 2.2. Now it reads:

Cooking experiments were conducted inside the tent by simulating the real Chinese cooking styles with different oils. To avoid the influences from burning of the fuel, an induction cooker was used in this study.

*Page 5, Line 20-24: If the CV-ACSM sampled $PM_{2.5}$ and the SV-AMS sampled $PM_1$, there might be a composition difference. When comparing the two (not only for COA but also for other OAs), please justify the conclusions with that in mind.*

*Also, the authors mentioned about less enhancement of f44 compared to Hu et al. 2018a. Can this be partially explained by the loading difference? I mean the loadings herein were 2 orders of magnitude greater than ambient OA concentrations. More volatile species may partition to the particle phase compared to Hu et al. 2018a as well as the other study that I mentioned earlier. Please indicate the difference of conditions when making the comparisons.*

We agree with the reviewer that there could be compositional difference between $PM_1$ and $PM_{2.5}$. In fact, a recent ambient study in north China indicated that the differences of primary OA between $PM_1$ and $PM_{2.5}$ were small even under high relative humidity conditions (Sun et al., 2020). In this work, the experiments were conducted during periods with relative humidity less than 60%, and the source spectra of primary OA

between $PM_1$ and $PM_{2.5}$ are not expected to be largely different. Most importantly, the size distributions of primary OA from AMS measurements showed that aerosol particles from burning different fuels were below 1 μm, supporting that the differences between $PM_1$ and $PM_{2.5}$ would not be important for this study.

Thanks for pointing out the loading effect on mass spectra. In fact, we compared $f_{44}$ of OA from burning different fuels under different mass loadings (Figure R1). Indeed, $f_{44}$ from biomass and wood burning overall showed relatively lower $f_{44}$ during periods with higher mass loadings, likely due to partitioning of more semi-volatile organic compounds under higher mass loadings. In contrast, higher $f_{44}$ for lower mass loadings could be due to the evaporation of semi-volatile organic compounds or rapid ageing of OA in the atmosphere. We clarified this in the revised manuscript with a new paragraph (see our response to the next comment).

"All COA spectral profiles measured by SV-AMS and CV-ACSM are highly similar ($R^2>0.89$, Fig. 5), and also resemble those previously resolved in ambient air during all seasons in Beijing (Fig. S4) despite the COA concentrations can have a difference of an order of magnitude. We also noticed slightly higher O/C ($f_{44}$) for COA under lower mass loadings, which were likely due to partitioning of more semi-volatile organics on particles during periods with higher mass loadings (Reyes-Villegas et al., 2018)."

"As shown in Figs. 3 and S6, all COA spectra of CV-ACSM are fairly stable and overall similar to those of SV-AMS ($R^2> 0.86$). Due to additional thermal decomposition in CV, the COA source spectra in CV showed slightly higher $f_{44}$ (2.4–3.7%) than that of SV-AMS (1.8–2.9%)(Hu et al., 2018a)."

[Figure]

Figure R1. The variations of $f_{44}$ and mass loadings of OA measured by SV-AMS in each experiment. The default RIE (1.4) and CE=1 were used.

*Page 10, Line 6-7: The water-soluble fraction of POA also depends on the atmospheric dilution of the primary sources. This study should discuss about the sampling OA loading levels and the possible change of solubility after atmospheric dilution. Would that change the order of solubility?*

Thanks for the reviewer's suggestion. The water-soluble fraction of POA in this study was estimated by the ratio of WSOC/OC with high mass loading. It should be noted that WSOA/OA ratio was also an indicator of water solubility in previous observations. The study of WSOC in Helsinki (Finland) and Paris (France) showed that 64% and 82% of the OC was water-soluble for wild land fires (Timonen et al., 2008) and wood burning (Sciare et al., 2011), respectively. By coupling a Particle-Into-Liquid-Sampler (PILS) and AMS, Xu et al. (2017) found that the average water solubility of BBOA was 75% with a large variation in southeastern America. The vertical distribution of WSOA sources in Beijing showed that 61 – 78% of BBOA was water-soluble at ground and 260 m level (Qiu et al., 2019). This discrepancy was likely due to the different biomass types and burning conditions. Similar to BBOA, the water-soluble fraction of COA also has a wide frequency distribution (8 – 40%)(Li et al., 2018). In additional, the water-solubility of COA estimated by combining online AMS and offline WSOA measurements was 19% at ground and 42% at 260 m in Beijing, this difference of COA water solubility was likely due to the ageing process associated with regional or vertical transport (Qiu et al., 2019). These results indicate that the order of water solubility of OA was unlikely changed due to the atmospheric dilution processes considering the large differences in water solubility for different primary OA.

We expanded the discussions in the revised manuscript. Now it reads:

"However, the spectral differences between water-soluble OA and the total OA can be substantial for both SV-AMS and CV-ACSM depending on water solubility which is in the order of BBOA > WBOA > COA > CCOA. Noted that the mass loadings of primary emissions in this experiment are much higher than those in ambient air, which could cause some differences in water solubility and subsequent spectral differences in WSOA."

*The loading-dependent OA composition also limits the application of the source profile directly in ME-2 (Page 10, Line 18-25). For example, if the source profile is obtained at 1000 ug m-3, the actual source in the atmosphere is indeed tens of ug m-3 after quick dilution. Many studies have shown the OA composition varied a lot at various loadings especially for combustion sources like BBOA and CCOA and hence may change the mass spectra. The authors should be clarify this complication and do not mislead users to use the source profiles without cautions.*

This is really a good point we didn't discuss before. Following the reviewer's suggestions, we further compared the source spectra of OA between low and high mass loadings, and also the changes in $f_{44}$ and $f_{60}$. We found that the mass spectra of COA and flaming combustion of coal are very stable across different mass loadings. Although the mass spectra of OA for the rest fuel burning are highly similar, the changes in $f_{44}$ and $f_{60}$ were also observed between low and high mass loadings. We then expanded the discussions on loading effects on mass spectra in the revised manuscript, and clarified the uncertainties that were caused by mass loadings. Now, the mass loadings for each burning experiment, and the comparisons of OA mass spectra between low and high mass loadings are all presented in supplementary (Figures S1 and S2, and Tables S1 and S2).

"The average mass loadings of OA during the burning and cooking experiments are nearly 2 order of magnitude of that in ambient air, indicating the negligible influences of background OA to our experiments. As shown in Table S1, the mass concentrations of OA measured by SV-AMS ranged from ~80 µg m$^{-3}$ to ~1370 µg m$^{-3}$ for different burning experiments by using a relative ionization efficiency of 1.4 and a collection efficiency of 1. Considering that the mass spectra of OA can have changes across different mass loadings due to the partitioning of semi-volatile organic compounds(Donahue et al., 2006; Shilling et al., 2009), we further checked the spectral differences between high and low mass loadings for SV-AMS and CV-ACSM (Tables S1 and S2, respectively). As indicated in Figures S1 and S2, the mass spectra of OA, and $f_{44}$, $f_{43}$, and $f_{60}$ from cooking and flaming combustion of coal are remarkably similar under low and high mass loadings, indicating that the mass spectra are relatively stable upon dilution or evaporation, and thus can be well used as constraints in source apportionment analysis. Although the mass spectra of OA for the rest burning, i.e., biomass burning, wood burning, and smoldering combustion of coal are also highly similar between low and high mass loadings, the ubiquitous increases in $f_{44}$ and corresponding decreases in $f_{60}$ were observed from high to low mass loadings. For instance, $f_{44}$ in SV-AMS was increased by 0.4 – 2% as the mass loading decreased by a factor of ~3, and $f_{60}$ showed a corresponding decrease by 0.1 – 0.9%. Similarly, $f_{44}$ in CV-ACSM was increased by 0.9 – 4.2% associated with a decrease in $f_{60}$ by 0.1 – 0.6% as OA mass loadings were decreased by a factor of ~3 – 4. Such results are consistent with previous studies that biomass burning OA can be rapidly aged in the atmosphere which is characterized by increases in $f_{44}$ and decreases in $f_{60}$ (Cubison et al., 2011; Morgan et al., 2020). Therefore, source apportionment of OA using the source spectra from biomass burning, wood burning and smoldering combustion of coal need to consider the mass loading effect and increase the variability uncertainties in $f_{44}$ and $f_{60}$. "

(a1) OM/OC=1.44; O/C=0.18; H/C=2.02; N/C=0.030
$C_xH_y^+$ $C_xH_yO_1^+$ $C_xH_yO_2^+$
$C_xH_yN_p^+$ $C_xH_yO_2N_p^+$
(b1) OM/OC=1.44; O/C=0.18; H/C=2.01; N/C=0.028

(a2) OM/OC=1.45; O/C=0.18; H/C=2.00; N/C=0.031
(b2) OM/OC=1.45; O/C=0.18; H/C=1.99; N/C=0.030

(a3) OM/OC=1.42; O/C=0.17; H/C=2.00; N/C=0.018
(b3) OM/OC=1.43; O/C=0.18; H/C=2.00; N/C=0.020

(a4) OM/OC=1.39; O/C=0.15; H/C=1.98; N/C=0.021
(b4) OM/OC=1.40; O/C=0.16; H/C=1.97; N/C=0.022

(a5) OM/OC=1.40; O/C=0.16; H/C=1.96; N/C=0.014
(b5) OM/OC=1.41; O/C=0.17; H/C=1.95; N/C=0.016

(a6) OM/OC=1.42; O/C=0.17; H/C=1.99; N/C=0.022
(b6) OM/OC=1.42; O/C=0.17; H/C=1.98; N/C=0.019

(a7) OM/OC=1.43; O/C=0.17; H/C=2.06; N/C=0.033
(b7) OM/OC=1.44; O/C=0.17; H/C=2.05; N/C=0.033

(a8) OM/OC=1.38; O/C=0.15; H/C=2.07; N/C=0.004
(b8) OM/OC=1.39; O/C=0.16; H/C=2.05; N/C=0.005

(a9) OM/OC=1.64; O/C=0.34; H/C=1.80; N/C=0.031
(b9) OM/OC=1.74; O/C=0.41; H/C=1.81; N/C=0.033

(a10) OM/OC=1.61; O/C=0.31; H/C=1.98; N/C=0.025
(b10) OM/OC=1.74; O/C=0.41; H/C=1.92; N/C=0.027

(a11) OM/OC=1.38; O/C=0.15; H/C=2.09; N/C=0.012
(b11) OM/OC=1.41; O/C=0.17; H/C=2.07; N/C=0.016

(a12) OM/OC=1.57; O/C=0.28; H/C=1.85; N/C=0.038
(b12) OM/OC=1.59; O/C=0.29; H/C=1.86; N/C=0.040

(a13) OM/OC=1.55; O/C=0.26; H/C=1.86; N/C=0.034
(b13) OM/OC=1.56; O/C=0.27; H/C=1.88; N/C=0.039

% of the total signal

$m/z_{high\ levels}$ (%)

S = 0.99; r² = 1.00
S = 1.00; r² = 1.00
S = 1.01; r² = 1.00
S = 1.00; r² = 1.00
S = 1.00; r² = 1.00
S = 1.00; r² = 1.00
S = 1.01; r² = 1.00
S = 1.01; r² = 1.00
S = 0.93; r² = 0.95
S = 0.96; r² = 0.87
S = 1.02; r² = 1.00
S = 1.00; r² = 1.00
S = 0.98; r² = 0.99

off*[continued]*

[Figure]

Figure S1. The mass spectral profiles of OA measured by SV-AMS at (a) high and (b) low levels from (1) stir-fried garlic with corn oil, (2) stir-fried celery with corn oil, (3) peanut oil, (4) bean oil, (5) sunflower oil, (6) blend oil, (7) lard oil and (8) barbecue (9) wheat, (10) corn, (11) bean, (12) rape, (13) cotton, (14) birchen, (15) pine tree, (16) poplar, (17) Chinese oak, (18) flaming combustion of brown coal, (19) smoldering combustion of brown coal, (20) flaming combustion of bituminous coal and (21) smoldering combustion of bituminous coal. The comparison of mass spectrum for each experiment is shown. The detailed descriptions of cooking and burning fuels are presented in Table S1.

[Figure]

Figure S2. The mass spectral profiles of OA measured by CV-ACSM at (a) high and (b) low levels from (1) stir-fried garlic with corn oil, (2) stir-fried celery with corn oil, (3) peanut oil, (4) bean oil, (5) sunflower oil, (6) blend oil, (7) lard oil and (8) barbecue (9) wheat, (10) corn, (11) bean, (12) rape, (13) cotton, (14) birchen, (15) pine tree, (16) poplar, (17) Chinese oak, (18) flaming combustion of brown coal, (19) smoldering combustion of brown coal, (20) flaming combustion of bituminous coal and (21) smoldering combustion of bituminous coal. The comparison of mass spectrum for each experiment is shown. The detailed descriptions of cooking and burning fuels are presented in Table S2.

**Table S1.** A summary of $f_{44}$, $f_{43}$, $f_{60}$ and OA concentration ($\mu g\ m^{-3}$) at high and low OA levels measured by SV-AMS in each experiment. The default RIE (1.4) and CE=1 were used.

| Fuels | OA | $f_{44}$ | $f_{43}$ | $f_{60}$ | OA | $f_{44}$ | $f_{43}$ | $f_{60}$ |
|---|---|---|---|---|---|---|---|---|
| | | High Conc. | | | | Low Conc. | | |
| CornOil1 | 503.9 | 0.027 | 0.078 | 0.006 | 382.9 | 0.029 | 0.077 | 0.006 |
| CornOil2 | 507.7 | 0.029 | 0.074 | 0.006 | 382.6 | 0.030 | 0.073 | 0.006 |
| Peanut | 564.3 | 0.021 | 0.077 | 0.008 | 388.8 | 0.023 | 0.077 | 0.008 |
| BeanOil | 485.0 | 0.023 | 0.068 | 0.006 | 338.3 | 0.024 | 0.069 | 0.006 |
| Sunflower | 624.0 | 0.017 | 0.070 | 0.007 | 388.3 | 0.019 | 0.070 | 0.007 |
| BlendOil | 666.9 | 0.020 | 0.070 | 0.006 | 313.8 | 0.023 | 0.070 | 0.006 |
| LardOil | 221.4 | 0.029 | 0.082 | 0.008 | 162.2 | 0.030 | 0.082 | 0.007 |
| BBQ | 421.7 | 0.014 | 0.095 | 0.010 | 150.4 | 0.018 | 0.094 | 0.009 |
| Wheat | 862.7 | 0.027 | 0.077 | 0.017 | 79.4 | 0.041 | 0.077 | 0.015 |
| Corn | 548.2 | 0.031 | 0.102 | 0.032 | 151.0 | 0.047 | 0.084 | 0.023 |
| Bean | 849.2 | 0.016 | 0.101 | 0.008 | 364.0 | 0.021 | 0.099 | 0.007 |
| Rape | 642.1 | 0.021 | 0.093 | 0.017 | 241.5 | 0.025 | 0.092 | 0.017 |
| Cotton | 742.0 | 0.023 | 0.084 | 0.019 | 264.9 | 0.028 | 0.084 | 0.016 |
| Birchen | 338.4 | 0.025 | 0.079 | 0.028 | 150.4 | 0.031 | 0.080 | 0.021 |
| Pine | 361.0 | 0.047 | 0.072 | 0.031 | 173.8 | 0.050 | 0.073 | 0.026 |
| Poplar | 1369.6 | 0.016 | 0.084 | 0.027 | 185.6 | 0.028 | 0.085 | 0.023 |
| Oak | 513.5 | 0.055 | 0.066 | 0.051 | 201.5 | 0.056 | 0.063 | 0.059 |
| BrCoalF | 154.2 | 0.029 | 0.042 | 0.003 | 154.4 | 0.029 | 0.041 | 0.003 |
| BrCoalS | 269.8 | 0.014 | 0.060 | 0.002 | 103.1 | 0.034 | 0.071 | 0.003 |
| BiCoalF | 376.4 | 0.006 | 0.091 | 0.001 | 339.4 | 0.005 | 0.092 | 0.001 |
| BiCoalS | 275.8 | 0.018 | 0.064 | 0.004 | 120.6 | 0.029 | 0.071 | 0.005 |

Note: CornOil1= stir-fried garlic with corn oil; CornOil2= stir-fried celery with corn oil; Peanut= stir-fried celery with peanut oil; Sunflower= stir-fried celery with sunflower oil; BeanOil= stir-fried celery with bean oil; BlendOil= stir-fried celery with blend oil; LardOil= stir-fried celery with lard oil; BBQ= barbecue; Wheat= dry wheat stalk burning; Corn= dry corn stalk burning; Bean= dry bean stalk burning; Rape= dry rape stalk burning; Cotton= dry cotton stalk burning; Birchen= dry birchen burning; Pine= dry pine tree burning; Poplar= dry poplar burning; Oak= dry Chinese oak burning; BrCoalF= brown coal combustion under flaming conditions; BrCoalS= brown coal combustion under smoldering conditions; BiCoalF= bituminous coal combustion under flaming conditions; BiCoalS= bituminous coal combustion under smoldering conditions.

**Table S2.** A summary of $f_{44}$, $f_{43}$, $f_{60}$ and OA concentration (µg m$^{-3}$) at high and low OA levels measured by CV-ACSM in each experiment. The default RIE (1.4) and CE=1 were used.

| Fuels | OA | $f_{44}$ | $f_{43}$ | $f_{60}$ | OA | $f_{44}$ | $f_{43}$ | $f_{60}$ |
|---|---|---|---|---|---|---|---|---|
| | High Conc. | | | | Low Conc. | | | |
| CornOil1 | 715.3 | 0.037 | 0.060 | 0.001 | 517.0 | 0.039 | 0.060 | 0.001 |
| CornOil2 | 714.4 | 0.033 | 0.056 | 0.001 | 265.3 | 0.039 | 0.055 | 0.000 |
| Peanut | 799.2 | 0.030 | 0.060 | 0.001 | 676.5 | 0.031 | 0.060 | 0.001 |
| BeanOil | 613.9 | 0.028 | 0.056 | 0.001 | 286.7 | 0.032 | 0.055 | 0.001 |
| Sunflower | 1064.2 | 0.023 | 0.053 | 0.001 | 730.2 | 0.025 | 0.053 | 0.001 |
| BlendOil | 927.0 | 0.024 | 0.056 | 0.001 | 829.7 | 0.024 | 0.056 | 0.001 |
| LardOil | 518.7 | 0.029 | 0.069 | 0.001 | 366.6 | 0.031 | 0.068 | 0.001 |
| BBQ | 290.4 | 0.030 | 0.081 | 0.004 | 102.5 | 0.038 | 0.079 | 0.003 |
| Wheat | 294.8 | 0.068 | 0.071 | 0.008 | 83.3 | 0.098 | 0.065 | 0.006 |
| Corn | 523.5 | 0.068 | 0.077 | 0.016 | 119.8 | 0.110 | 0.064 | 0.012 |
| Bean | 893.0 | 0.032 | 0.095 | 0.004 | 279.7 | 0.046 | 0.088 | 0.004 |
| Rape | 683.8 | 0.035 | 0.095 | 0.012 | 172.8 | 0.048 | 0.090 | 0.011 |
| Cotton | 615.2 | 0.044 | 0.082 | 0.015 | 188.1 | 0.060 | 0.077 | 0.012 |
| Birchen | 558.3 | 0.042 | 0.068 | 0.022 | 206.8 | 0.058 | 0.067 | 0.016 |
| Pine | 402.8 | 0.068 | 0.070 | 0.029 | 107.1 | 0.084 | 0.069 | 0.023 |
| Poplar | 616.4 | 0.032 | 0.087 | 0.023 | 104.8 | 0.060 | 0.081 | 0.018 |
| Oak | 485.9 | 0.091 | 0.057 | 0.039 | 133.4 | 0.100 | 0.054 | 0.049 |
| BrCoalF | 121.6 | 0.057 | 0.034 | 0.002 | 82.1 | 0.061 | 0.039 | 0.002 |
| BrCoalS | 269.4 | 0.031 | 0.059 | 0.001 | 104.8 | 0.054 | 0.063 | 0.001 |
| BiCoalF | 334.9 | 0.015 | 0.094 | 0.001 | 241.4 | 0.016 | 0.096 | 0.001 |
| BiCoalS | 276.8 | 0.031 | 0.061 | 0.003 | 101.4 | 0.061 | 0.068 | 0.003 |

Note: CornOil1= stir-fried garlic with corn oil; CornOil2= stir-fried celery with corn oil; Peanut= stir-fried celery with peanut oil; Sunflower= stir-fried celery with sunflower oil; BeanOil= stir-fried celery with bean oil; BlendOil= stir-fried celery with blend oil; LardOil= stir-fried celery with lard oil; BBQ= barbecue; Wheat= dry wheat stalk burning; Corn= dry corn stalk burning; Bean= dry bean stalk burning; Rape= dry rape stalk burning; Cotton= dry cotton stalk burning; Birchen= dry birchen burning; Pine= dry pine tree burning; Poplar= dry poplar burning; Oak= dry Chinese oak burning; BrCoalF= brown coal combustion under flaming conditions; BrCoalS= brown coal combustion under smoldering conditions; BiCoalF= bituminous coal combustion under flaming conditions; BiCoalS= bituminous coal combustion under smoldering conditions.

*Technical Remarks:*

*"/" in "m/z" should not be italized.*

Changed

References

Faber, P., Drewnick, F., Veres, P. R., Williams, J., and Borrmann, S.: Anthropogenic sources of aerosol particles in a football stadium: Real-time characterization of emissions from cigarette smoking, cooking, hand flares, and color smoke bombs by high-resolution aerosol mass spectrometry, Atmos. Environ., 77, 1043-1051, 10.1016/j.atmosenv.2013.05.072, 2013.

Fang, Z., Deng, W., Zhang, Y. L., Ding, X., Tang, M. J., Liu, T. Y., Hu, Q. H., Zhu, M., Wang, Z. Y., Yang, W. Q., Huang, Z. H., Song, W., Bi, X. H., Chen, J. M., Sun, Y. L., George, C., and Wang, X. M.: Open

burning of rice, corn and wheat straws: primary emissions, photochemical aging, and secondary organic aerosol formation, Atmos. Chem. Phys., 17, 14821-14839, 10.5194/acp-17-14821-2017, 2017.

Jolleys, M. D., Coe, H., McFiggans, G., Taylor, J. W., O'Shea, S. J., Le Breton, M., Bauguitte, S. J. B., Moller, S., Di Carlo, P., Aruffo, E., Palmer, P. I., Lee, J. D., Percival, C. J., and Gallagher, M. W.: Properties and evolution of biomass burning organic aerosol from Canadian boreal forest fires, Atmos. Chem. Phys., 15, 3077-3095, 10.5194/acp-15-3077-2015, 2015.

Kaltsonoudis, C., Kostenidou, E., Louvaris, E., Psichoudaki, M., Tsiligiannis, E., Florou, K., Liangou, A., and Pandis, S. N.: Characterization of fresh and aged organic aerosol emissions from meat charbroiling, Atmos. Chem. Phys., 17, 7143-7155, 10.5194/acp-17-7143-2017, 2017.

Li, Y. C., Qiu, J. Q., Shu, M., Ho, S. S. H., Cao, J. J., Wang, G. H., Wang, X. X., and Zhao, X. Q.: Characteristics of polycyclic aromatic hydrocarbons in PM2.5 emitted from different cooking activities in China, Environ. Sci. Pollut. Res., 25, 4750-4760, 10.1007/s11356-017-0603-0, 2018.

Liu, T., Wang, Z., Wang, X., and Chan, C. K.: Primary and secondary organic aerosol from heated cooking oil emissions, Atmos. Chem. Phys., 18, 11363-11374, 10.5194/acp-18-11363-2018, 2018.

Liu, T. Y., Li, Z. J., Chan, M. N., and Chan, C. K.: Formation of secondary organic aerosols from gas-phase emissions of heated cooking oils, Atmos. Chem. Phys., 17, 7333-7344, 10.5194/acp-17-7333-2017, 2017.

McLafferty, F. W., and Turecek, F.: Interpretation of Mass Spectra, University Science Books, Mill Valley, California, 1993.

Qiu, Y., Xie, Q., Wang, J., Xu, W., Li, L., Wang, Q., Zhao, J., Chen, Y., Chen, Y., Wu, Y., Du, W., Zhou, W., Lee, J., Zhao, C., Ge, X., Fu, P., Wang, Z., Worsnop, D. R., and Sun, Y.: Vertical Characterization and Source Apportionment of Water-Soluble Organic Aerosol with High-resolution Aerosol Mass Spectrometry in Beijing, China, ACS Earth and Space Chemistry, 3, 273-284, 10.1021/acsearthspacechem.8b00155, 2019.

Reyes-Villegas, E., Bannan, T., Le Breton, M., Mehra, A., Priestley, M., Percival, C., Coe, H., and Allan, J. D.: Online Chemical Characterization of Food-Cooking Organic Aerosols: Implications for Source Apportionment, Environ. Sci. Technol., 52, 5308-5318, 10.1021/acs.est.7b06278, 2018.

Sciare, J., d'Argouges, O., Sarda-Estève, R., Gaimoz, C., Dolgorouky, C., Bonnaire, N., Favez, O., Bonsang, B., and Gros, V.: Large contribution of water-insoluble secondary organic aerosols in the region of Paris (France) during wintertime, J. Geophys. Res., doi:10.1029/2011JD015756, 2011.

Sun, Y., He, Y., Kuang, Y., Xu, W., Song, S., Ma, N., Tao, J., Cheng, P., Wu, C., Su, H., Cheng, Y., Xie, C., Chen, C., Lei, L., Qiu, Y., Fu, P., Croteau, P., and Worsnop, D. R.: Chemical Differences Between PM1 and PM2.5 in Highly Polluted Environment and Implications in Air Pollution Studies, Geophys. Res. Lett., 47, e2019GL086288, 10.1029/2019gl086288, 2020.

Timonen, H., Saarikoski, S., Tolonen-Kivimäki, O., Aurela, M., Saarnio, K., Petäjä, T., Aalto, P., Kulmala, M., Pakkanen, T., and Hillamo, R.: Size distributions, sources and source areas of water-soluble organic carbon in urban background air, 2008.

Xu, L., Guo, H., Weber, R. J., and Ng, N. L.: Chemical Characterization of Water-Soluble Organic Aerosol in Contrasting Rural and Urban Environments in the Southeastern United States, Environ. Sci. Technol., 51, 78-88, 10.1021/acs.est.6b05002, 2017.

Zheng, Y., Cheng, X., Liao, K., Li, Y., Li, Y., Huang, R. J., Hu, W., Liu, Y., Zhu, T., Chen, S., Zeng, L., Worsnop, D. R., and Chen, Q.: Characterization of Anthropogenic Organic Aerosols by TOF-ACSM with the New Capture Vaporizer, Atmos. Meas. Tech. Discuss., 2020, 1-24, 10.5194/amt-2019-449, 2020.